# Plant Growth-Promoting Rhizobacteria for Sustainable Agricultural Production

**DOI:** 10.3390/microorganisms11041088

**Published:** 2023-04-21

**Authors:** Luana Alves de Andrade, Carlos Henrique Barbosa Santos, Edvan Teciano Frezarin, Luziane Ramos Sales, Everlon Cid Rigobelo

**Affiliations:** Agricultural and Livestock Microbiology Graduate Program, School of Agricultural and Veterinarian Sciences, São Paulo State University (UNESP), São Paulo 14884-900, Brazil; luanaalvesdeandrade2018@gmail.com (L.A.d.A.); chsantos@outlook.com (C.H.B.S.); edvan-frezarin@hotmail.com (E.T.F.); luziane.sales@unesp.br (L.R.S.)

**Keywords:** food production, nitrogen fixation, phosphorus solubilization, siderophores

## Abstract

Rhizosheric bacteria with several abilities related to plant growth and health have been denominated Plant Growth-Promoting Rhizobacteria (PGPR). PGPR promote plant growth through several modes of action, be it directly or indirectly. The benefits provided by these bacteria can include increased nutrient availability, phytohormone production, shoot and root development, protection against several phytopathogens, and reduced diseases. Additionally, PGPR can help plants to withstand abiotic stresses such as salinity and drought and produce enzymes that detoxify plants from heavy metals. PGPR have become an important strategy in sustainable agriculture due to the possibility of reducing synthetic fertilizers and pesticides, promoting plant growth and health, and enhancing soil quality. There are many studies related to PGPR in the literature. However, this review highlights the studies that used PGPR for sustainable production in a practical way, making it possible to reduce the use of fertilizers such as phosphorus and nitrogen and fungicides, and to improve nutrient uptake. This review addresses topics such as unconventional fertilizers, seed microbiome for rhizospheric colonization, rhizospheric microorganisms, nitrogen fixation for reducing chemical fertilizers, phosphorus solubilizing and mineralizing, and siderophore and phytohormone production for reducing the use of fungicides and pesticides for sustainable agriculture.

## 1. Introduction

Plant growth-promoting rhizobacteria (PGPR) are free-living bacteria that colonize plant roots and promote plant growth. PGPR may promote plant growth by using their own metabolism (solubilizing phosphates, producing hormones, or fixing nitrogen), by directly affecting the plant metabolism (increasing the uptake of water and minerals), enhancing root development, increasing the enzymatic activity of the plant, by “helping” other beneficial microorganisms to enhance their action on the plant, or by suppressing plant pathogens [1,2,3]. They protect plants indirectly by competing with pathogens for scarce nutrients, biocontrolling pathogens by producing aseptic-activity compounds, synthesising fungal cell wall lysing enzymes, and inducing systemic responses in host plants. PGPR may help plants to thrive under abiotic stress by improving the plant fitness, stress tolerance, and pollution remediation. Additional data and greater knowledge of bacterial features driving plant-growth promotion might motivate and stir the development of creative solutions utilizing PGPR in highly changeable environmental and climatological settings [4].

## 2. The Use of Unconventional Fertilizers

Agricultural production’s challenge is constantly increasing agricultural production and improving its quality, processing, and storage. Plant cultivation requires increasing the yield through the effective use of mineral fertilizers. It is well known that applying mineral fertilizers of the optimal standards to the soil is important for improving soil condition, increasing its fertility, and inducing the productivity of crops [5]. Many agronomic practices may need to be adjusted to maximize food production yield and quality. Thus, agronomic packages must be continuously modified. In recent years, many investigators have applied biofertilizers to minimize the environmental pollution which results from mineral fertilizers and also to reduce their costs [6]. The use of waste to produce liquid fertilizers in terms of sustainable agriculture is a promising practice that could help food production. Pajura et al. [7] wrote a fascinating review discussing the challenge faced by the fertilizer industry to produce sufficient nutrients for plant growth using more energy-efficient and environmentally friendly methods. The production of liquid fertilizers from waste materials that exhibit fertilizer properties is proposed as a solution to reduce the exploitation of natural resources and implement elements of a circular economy. The review highlights the current regulations in Poland and the European Union aimed at promoting a circular economy and the need to obtain fertilizers containing valuable plant nutrients from organic waste or recycled materials. The review also identifies the waste materials used as substrates to produce fertilizers and their important chemical properties for plant growth and development. The study also emphasizes the importance of this line of research and the need to look for other waste groups for reuse within the circular economy framework. Luo et al. [8] investigated the effects of a new organic–inorganic-compound fertilizer on the growth, yield formation, and aroma biosynthesis of fragrant rice. The fertilizer was made with organic matter, urea, superphosphate, potassium chloride, zinc sulfate, and lanthanum chloride. The study was conducted over four years, and three treatments were used: no fertilizer, traditional fertilizer, and the new organic–inorganic-compound fertilizer. The results showed that the new fertilizer significantly increased the grain yield, effective panicle number, seed-setting rate, chlorophyll content, net photosynthetic rate, aboveground biomass, and 2-acetyl-1-pyrroline content in fragrant rice compared to other treatments. The study suggested that the new fertilizer could achieve a high yield and grain content in fragrant rice production. Dhawi et al. [9] discussed the benefits of using plant growth-promoting microorganisms (PGPMs) in hydroponics and vertical farming systems. The controlled environment in these systems allows for maximizing the use of PGPMs. The authors recommend a synchronized PGPM treatment using a biostimulant extract added to the hydroponic medium while also pre-treating seeds or seedlings with a microbial suspension for aquaponic and aeroponic systems. The global market for vertical farming is predicted to reach more than USD 10.02 billion by 2027 due to the sustainable use of space, reduction in water use, lack of pesticides, and the implementation of user-friendly technology for environmental control and harvesting.

## 3. Seed Microbiome for Rhizospheric Colonization

Seeds contain several endophytic microorganisms, especially bacteria, that the plant selects due to their many benefits [10,11,12]. Initially, these microorganisms colonize the rhizosphere, then they inhabit the plant tissue as endophytes and later are transferred to the seeds [13]. In many sources, endophytes play a crucial role in seed germination, conservation, and development, and these microorganisms are found in the soil. The results suggest that the plant selects these microorganisms from the rhizosphere because of the benefits they provide to the plant to guarantee their presence when the seeds are planted [14]. Like any plant organ, seed endophyte colonization relies on different chemical compositions. Moreover, the defence mechanism hinders high population density inside the plant organ, which could provoke an infection due to quorum sensing [15,16].

Endophyte seeds are transmitted vertically from the roots to the stem. Unlike distant rhizobacteria, bacterial endophytes interact tightly with the growing embryo in the germination stage. Seed endophytes can help the seedlings to grow and develop by solubilizing potassium and phosphorus, generating the growth of several hormone levels, such as cytokinin and auxin, and fixing N. Endophyte seeds also benefit plants with resistance to biotic and abiotic stressors and better overall fitness [17,18]. Pal et al. [19] isolated twenty-three microbial endophytes (bacteria) from maize seeds, and 70% of them presented the ability to synthesize auxin, 74% demonstrated several abilities, including the ability to solubilize phosphate, and all the isolates showed the ability to fix nitrogen. In their study, several isolates showed antagonistic action against phytopathogenic fungi such as *Fusarium* sp. and *Rhizoctonia solani*, indicating their potential for biocontrol. Jana et al. [20] identified seed rice bacteria such as *Bacillus* sp., *Citrobacter* sp., *Flavobacterium* sp., and *Pantoa* sp., which had not previously been isolated from rice cultivar seeds. *Citrobacter* produced the most indole acetic acid (IAA), gibberellin, and hydrogen cyanide (HCN) among the isolates tested. At the same time, *Pantoa* demonstrated the maximum efficiency for phosphate and potassium solubilization and ammonia production. These findings imply that the endophytes of separated seeds can aid plant growth and development while assisting host plants in their battle against several phytopathogens, such as fungi and bacteria, in sustainable production.

## 4. Rhizospheric Microorganisms

The portion of the soil close to the roots that suffers nutritional interference from the roots is named rhizosphere [18,21]. Plants perform photosynthesis, and depending on the plant species, they invest 10 to 40% of their photosynthetic metabolites in the rhizosphere through rhizodeposition. Through rhizodeposition, the rhizospheric soil is fertilized and enriched with nutrients, amino acids, and organic energetic molecules such as carbohydrates [22,23]. Fertilization of the rhizosphere exerts a significant influence and changes the soil microbiota near the roots. Plants modulate rhizospheric microorganisms through plant physiological factors that govern plant–microorganisms interactions and by the composition of their exudates [24]. The composition of the microorganisms in the roots is selected, and the population of the root microbiome takes place in two stages. The first stage is rhizosphere colonization, accomplished by a subgroup of microorganisms from the nonrhizosphere soil and bulk soil. In the second stage, the phyllosphere and endosphere are colonized by a subset of microorganisms from the rhizosphere [25].

Several factors affect the halobionts of plants and modulate the microbiome composition. A holobiont is a set of a plant’s genome with its microbiome’s genome. Despite the fact that unlike plants grown under different conditions, they have the same groups of microorganisms. The group of microorganisms that persists in different plants is called the core microbiome. The core microbiome is formed by factors common to different plants. On the other hand, plant-specific factors result in associations with microorganisms that are not part of the core microbiome. Rhizospheric microorganisms obtain nutrients and energy molecules in the rhizosphere from rhizodeposition and border cells eliminated by the roots. In this sense, the larger is the root volume the greater is the deposition and elimination of these molecules and, consequently, the energy and nutrient availability to the microbial population of the rhizosphere. On the other hand, a more significant development of the plant’s shoot increases its photosynthetic efficiency and the generation of energy molecules [26]. In this sense, these microorganisms are called phytostimulants. They synthesize phytohormones while gaining excellent benefits for themselves and promoting shoot and root development [27]. Phytohormones play an essential role in increasing microorganism survival by cancelling plant defence against themselves [27,28].

Another way in which PGPR can promote plant growth is by inducing systemic resistance (ISR) and systemic acquired resistance (SAR) in plants. These are defence mechanisms that plants use to protect themselves against pathogenic bacteria, viruses, and fungi [29]. ISR is triggered by non-pathogenic microorganisms and starts in the root, extending to the shoot [30]. This defence response is dependent on ethylene and jasmonic acid signalling in the plant. In contrast, SAR is typically activated by necrotic pathogenic bacteria, and the signalling molecules that play important role in plant growth and defence [31]. 

## 5. Microorganisms Skills to Promote Plant Growth

As the population continues growing, commodity commercialization has been increased, while the agricultural lands have been reduced due to soil degradation. The food production sector has suffered high pressure to maintain its productivity [4]. This particular situation requires the utilization of chemical fertilizers and pesticides. The excessive use of these chemicals may provoke problems for the environment and human diseases. Current agriculture needs alternatives that reduce cost production, environmental impact, and dependence on input reduction without reducing productivity. In this way, microbial agents, especially microorganisms, which show several abilities related to plant growth, can be used as a helpful alternative [18,32,33].

The microorganisms might have a helpful, harmful, or neutral interaction with the host plant. Microorganisms that provide several benefits to the plant have strong potential for application as biopesticides and biofertilizers sustaining and improving crop protection and output. The challenge is making farmers worldwide utilize biofertilizers and biological control agents to reduce the use of chemical fertilizers and pesticides [4,34]. Microorganisms, especially plant growth-promoting microorganisms, may interact with several crop plants, improving their assistance to plant growth and development to resist pathogen attack and to promote their development. Several metabolites produced by microorganisms have been recognized, for commercial application, because of their helpful abilities to promote plant growth, mass production, biocontrol efficiency, and adequate formulation [4,26]. Various biocomplexes have been identified, such as biopesticides and biofertilizers, that can safeguard plants against both biotic and abiotic stresses. They accomplish this by generating plant growth regulators and siderophores, improving nutrient absorption, enhancing yield, and producing antagonistic compounds such as hydrolytic enzymes, antibiotics, volatile compounds, and hydrogen cyanides [22].

### 5.1. Nitrogen Fixation for Reducing Chemical Fertilizers

Nitrogen (N) is essential for numerous crop production processes [35]. Their grain productivity significantly depends on N input. As food demand continues growing, the necessity of nitrogen has been increasing too [36]. Rice, maize, potatoes, and wheat are crops that benefit from increased application of nitrogen fertilizer to improve their productivity. However, despite the relatively low efficiency in using nitrogen, which is mostly due to processes such as ammonia volatilization, N leaching, and denitrification, rice cultivation alone accounts for 21–25% of the world’s total N fertilizer application. This is because nitrogen is the most essential macronutrient in plant physiology [37]. Nitrogen is the most crucial nutrient required in large quantities for maize production as it plays a significant role in the formation of amino acids, chlorophyll, adenosine triphosphate (ATP), and nucleic acids. Therefore, increasing the application of nitrogen fertilizer for maize cultivation is directly proportional to its yield potential [38]. Biological nitrogen fixation (BNF) is one option for reducing reliance on chemical nitrogen fertilizers. Moreover, BNF accounts for more than 60% of the fixed N on Earth. As a result, maximizing BNF in agriculture is becoming increasingly important to fulfil the expanding global population’s demand for food production. This optimization will need a thorough understanding of the various nitrogen-fixing bacteria and their processes [39]. Jia et al. [40] identified and used the nitrogen-fixing bacteria *Kosakonia radicincitans* from *Pennisetum giganteum*. These researchers discovered a 25% reduction in chemical fertilizer combined with microorganisms. This combination increased plant height, weight, chlorophyll content, soluble protein content, soluble sugar content, vitamin C content, alkali hydrolysed nitrogen content, and accessible phosphorus content. Song et al. [41] evaluated urea reduction for two years as artificial N fertilizers replaced cyanobacteria *Anabaena azotica* in rice production. The results revealed that substituting 50% urea for the cyanobacteria did not appreciably reduce rice output. Additionally, the data revealed that traditional fertilization resulted in the highest N loss, whereas substituting *A. azotica* for partial urea greatly decreased NH_4^+^_
^−^N and NO_3_− leaching losses. In addition, replacing 50% of urea with *A. azotica* towards the end of the rice season can result in better retention of soil nitrogen compared to conventional fertilizers. This is because *A. azotica* has the ability to intercept, fix, and delay the release of nitrogen, which can greatly benefit the N cycling dynamics of the soil, leading to significant reductions in N leaching. Several studies have been conducted to evaluate the effects of diazotroph microorganisms such as *A. azotica* on maize yield. Tapia-Garcia [42] discovered the most common nitrogen-fixing endophyte, *Burkholderia*, associated with maize, has been a significant breakthrough. Recent studies have confirmed that these isolates can densely colonize maize tissues, leading to a significant increase in production. Sheoran [43] conducted research to investigate maize–endophyte relationships and their influence on maize output under both laboratory and field circumstances, and the association of *Klebsiella pneumoniae* with *Herbaspirillum seropedicae* endophytes resulted in a considerable increase in yield. Pandey et al. [44] experimented with local maize cultivars using *Azotobacter chroococcum* and *Azospirillum brasilense* strains. Under tropical circumstances, they discovered a considerable 1–1.5-fold increase in maize productivity.

### 5.2. Phosphorus Solubilizing and Mineralizing and Siderophore Production

Phosphorus (P) is a crucial macronutrient required for the growth and metabolism of plants. However, when added to soil, P is quickly immobilized by metal cations (such as Al, Fe, and Ca) or bound to mineral surfaces, which results in limited P availability for plant uptake [45]. Phosphates participate in physiological and biochemical processes, such as photosynthesis, root and stem development, flower and seed formation, crop maturation, nitrogen fixation by legumes, and plant disease resistance. Phosphates are one of the most important factors limiting agricultural production [46,47].

Wan et al. [48] conducted research to evaluate the potential of eight bacterial genera, including *Acinetobacter*, *Pseudomonas*, *Massilia*, *Bacillus*, *Arthrobacter*, *Stenotrophomonas*, *Ochrobactrum*, and *Cupriavidus*, to solubilize phosphorus. The results indicated that *Acinetobacter* exhibited a remarkable ability to solubilize phosphorus, making it a promising candidate for enhancing soil fertility and quality [46]. Liu et al. [49] have shown that phosphorus solubilizing bacteria have the ability to secrete small molecular organic acids that can dissolve inorganic phosphorus, which in turn can alter soil properties and indirectly influence the microbial community in the rhizosphere. Pantigoso et al. [50] investigated the effectiveness of bacteria such as *Enterobacter cloacae*, *Pseudomonas pseudoalcaligenes*, and *Bacillus thuringiensis* in solubilizing plant-unavailable P in either inorganic (calcium phosphate) or organic (phytin) forms. The study found that threonine played a vital role in promoting bacterial solubilization and plant uptake of various nutrients. The authors also suggested that specialized compounds exuded by these bacteria could be a promising approach to unlock existing phosphorus reservoirs in croplands. Kour et al. [51] evaluated the ability of various genera of plant growth-promoting bacteria, including *Bacillus*, *Enterobacter*, *Pseudomonas*, *Staphylococcus*, *Acinetobacter*, *Klebsiella*, and *Proteus*, to solubilize a significant amount of phosphorus from soil samples collected from the Lesser Himalayas ecosystem. The results indicated that these bacteria demonstrated a remarkable capacity to solubilize phosphorus, suggesting their potential for enhancing soil fertility and plant growth. Thus, these bacteria could be used for reducing the amount of phosphorus fertilizers. 

Iron (Fe) is another essential mineral for plants. It typically occurs as Fe^3+^ and Fe^2+^. Iron in soil can take different forms, such as insoluble hydroxides and oxyhydroxides in aerobic environments, making it unavailable for plant absorption. However, rhizospheric bacteria secrete siderophores, which are low molecular weight iron chelators with a high affinity for complex iron. Several plant growth-promoting rhizobacteria (PGPR) species, including *Enterobacter*, *Pseudomonas*, *Azotobacter*, *Bacillus*, *Serratia*, and *Rhizobium*, produce siderophores that can be either extracellular or intracellular, water-soluble, able to solubilize iron from minerals or organic molecules under iron-limiting conditions, and capable of forming stable complexes with heavy metals and radioactive particles. These siderophore-producing PGPR strains are beneficial for enhancing plant growth and mitigating heavy metal toxicity in contaminated soils [52]. This capacity indirectly aids the host plant in alleviating soil-caused heavy metal stress. Plants assimilate iron from siderophores through a variety of mechanisms, including chelating and releasing iron, direct absorption of siderophore–iron complexes, and ligand exchange. Siderophores play a dual function in iron sequestration and mitigation of plant stress induced by heavy metals. *Pseudomonads* generate siderophores with a high affinity for ferric ions [53]. It has been demonstrated that the formation of siderophores by biocontrol pseudomonads suppresses phytopathogens such as *Aspergillus*, *Fusarium*, and *Pythium* species [54]. Pyoverdine, a siderophore generated by pseudomonads, has been shown to reduce *Fusarium-oxysporum*-caused potato wilt [55]. Peanuts and maize also inhibited the phytopathogens *Fusarium moniliforme*, *Fusarium graminearum*, and *Macrophomina phaseolina* [55]. As a result, a lack of iron intake may be growth-limiting. In soil, Fe is mostly unavailable in a ferric oxidation state (Fe^3+^), and it is anticipated to generate insoluble hydroxides with extremely low solubility constants, rendering it inaccessible to plants and rhizospheric bacteria [56]. On the other hand, the ferrous (Fe^2+^) state is substantially more soluble and accessible to plants, but in the environment, it easily oxidizes into Fe^3+^, precipitating [57]. To persist in iron-deficient environments, most microorganisms have evolved a high-affinity iron (Fe^3+^) absorption mechanism involving siderophores and low-molecular-mass organic molecules (iron chelators). Siderophores function as iron solubilizers by combining with Fe^3+^ on bacterial membranes and then reducing it to Fe^2+^, making it accessible to both themselves and plants in iron-deficient environments [58]. The siderophores are ejected and recycled for iron transport after they are liberated inside the cells. According to Sultana et al. [59] both soil salinity and Fe deficiency negatively impact plant stem and root development, photosynthesis, transpiration rates, chlorophyll concentration, and stomatal conductance. Searching for potential siderophore-producing, salt-tolerant PGPR could be useful for cultivating salinity-affected regions without the use of transgenic organisms. These plant growth-promoting bacteria, *Gluconacetobacter diazotrophicus* and *Azospirillum brasilense,* were evaluated in [60]. In their absence, iron can create hydroxamate and catechol-type siderophores, which chelate the metal and facilitate its absorption. Iron, which is involved in physiological processes and is a component of several essential compounds, is required by plants. Using the growth index, leaf and root area, greenness index, total soluble phenolic compounds, and total iron content, this study sought to assess the contribution of two siderophore-generating bacteria to iron nutrition for strawberry plants. Strawberry plants were grown hydroponically in Hoagland nutrient solution with a 16 h photoperiod, iron sources were altered, and each bacterium was introduced. On day 60, the treatments with decreased iron had the maximum growth index, root area, greenness index, and iron content, whereas those without iron addition had the lowest values. The study showed similar results between plants inoculated with bacteria and those exposed to oxidized iron, compared to the untreated plants and reduced iron. After 30 days, infected plants showed decreased levels of phenolic compounds, which were higher in the iron-free and uninoculated treatments. Iron-deficient plants with bacterial inoculation had low concentrations of phenolic compounds. The study also found that *G. diazotrophicus* and *A. brasilense* siderophores can enhance iron nutrition in hydroponically grown strawberry plants. Specifically, hydroxamates were more effective than catechols in providing iron to the plants. According to Ferreira [61], another important aspect to consider with siderophores in the environment is their potential for abiotic degradation, which can occur through hydrolysis and/or oxidation mechanisms. For siderophores containing hydroxamate moieties, hydrolysis can lead to the formation of hydroxylamine groups, which in turn can reduce Fe^3+^ to Fe^2+^. In laboratory studies, hydrolysed products from coprogen (a trihydroxamate siderophore) were found to be effective iron transporters for cucumber and maize plants. All of these facts point to a putative siderophore usage strategy in which the presence of “sacrificial” moieties may aid in reducing, dissolving, and delivering iron to (micro)organisms. The processes of siderophore breakdown and mineral dissolution have also been altered by sunshine exposure. The presence of chelated Fe, as well as the kind of siderophore, might cause distinct effects. According to Ghazi [62], another critical aspect to consider regarding siderophores in the environment is their susceptibility to abiotic degradation, which can occur through hydrolysis and oxidation mechanisms. In the case of siderophores containing hydroxamate moieties, hydrolysis can lead to the formation of hydroxylamine groups, which can then reduce Fe^3+^ to Fe^2+^. In laboratory studies, researchers found that hydrolysed products from coprogen, a trihydroxamate siderophore, were effective in transporting iron to cucumber and maize plants. Kumar [63] evaluated the impact of four organophosphate pesticides, namely, acephate, glyphosate, monocrotophos, and phorate, on soil microorganisms that produce siderophores or plant growth-promoting rhizobacteria (PGPR). Five siderophore-producing soil microorganisms, namely, *Rhizobium leguminosarum*, *Pseudomonas fluorescens*, *Azotobacter vinelandii*, *Bacillus brevis,* and *Salmonella typhimurium*, were tested both individually and in combination with the pesticides. Results of the siderophore generation test showed a dose-dependent impact, and the impacts of the pesticide mixtures were more substantial than those of the individual pesticides. The overall sequence of unfavorable effects on siderophore synthesis caused by the four pesticides was phorate, acephate, monocrotophos, and glyphosate, which was consistent with the pesticides’ toxicity levels. The study also found that the pesticides had the least effect on the PGPR strain *Pseudomonas fluorescens* (13–66%), whereas *Salmonella typhimurium* had the least effect (20–75%). Pesticides had the following unfavorable effects on PGPR strains: *Bacillus brevis* (19–80%), *Salmonella typhimurium* (20–75%), *Rhizobium leguminosarum* (21–72%), *Azotobacter vinelandii* (22–81%), and *Pseudomonas fluorescens* (13–66%). Additionally, the combination of glycine and monocrotophos had little or no negative impact on the PGPR strains.

### 5.3. Phytohormone Production for Reducing Fungicides and Pesticides

The ability of plant growth-promoting bacteria (PGPB) to produce phytohormones such as indole-3-acetic acid (IAA), cytokinin, and gibberellin can significantly affect the hormonal balance of plants. IAA, in particular, can directly influence the plant’s endogenous reservoir of auxin. The overall effect of bacteria-produced IAA on root development depends on the total quantity of IAA available to the plant and the plant’s sensitivity to the hormone, which may result in either a positive or negative effect. Bacterial auxin at modest concentrations may stimulate growth. At optimal levels of endogenous auxin, introducing auxin from a PGPR source may inhibit or suppress plant growth. The stimulation of lateral and adventitious root growth by bacterial IAA improves nutrient uptake efficiency. In addition, it promotes root exudation. This cycle proceeds as increased root exudation leads to increased bacterial proliferation. However, it is also true that IAA production alone cannot explain a plant’s ability to promote growth because it indirectly inhibits root elongation [64]. Khan et al. [65] have determined that the excessive use of fungicides in agriculture can lead to a significant accumulation of active residues in the soil, resulting in negative impacts on crop health and yield. To investigate the potential positive interactions of radish plants with fungicide-tolerant plant growth-promoting rhizobacteria, the response of *Raphanus sativus* (white radish) to fungicides in the soil was analysed. PGPR were isolated from cabbage and mustard rhizospheres. The isolates of fungicide-tolerant PGPR were closely related to *Pseudomonas* spp. based on their morphological and biochemical characteristics, as well as their fragmentary 16S rRNA gene sequences. This PGPR was resistant to high concentrations of fungicides, including carbendazim and hexaconazole. Even when exposed to fungicides, bacterial isolates generated plant growth stimulants, although fungicides caused surface morphological distortion and changes in membrane permeability, as evidenced by microscopic examinations. Fungicides significantly impacted the germination efficiency, growth, and physiological development of *R. sativus*, and these effects were alleviated when the plants were inoculated with PGPR isolates. The application of carbendazim resulted in a reduction in whole dry biomass, a 54% decrease in whole plant length, a decline in total chlorophyll, a drop in protein content, and a 29% decrease in carotenoid synthesis. However, using isolate on white radish cultivated in soil amended with carbendazim improved plant growth and development, increasing whole plant dry weight by 10%, overall plant length, and total chlorophyll content. Likewise, the isolate enhanced plant performance by reducing proline, malondialdehyde, ascorbate peroxidase, catalase, and glutathione reductase (4%). Incorporating both isolates can be a practical approach for remediating fungicide-contaminated soil, as well as enhancing radish plant growth while reducing fungicide inputs. Enhazi [66] conducted a study to assess the efficacy of plant growth-promoting (PGP) rhizobacterial strains that are resistant to pesticide toxicity. The researchers isolated *Pseudomonas* sp. From the rhizosphere of *Vigna radiata* (L.) which produced various growth-regulating (GR) substances including indole-3-acetic acid, 1-aminocyclopropane ammonia-1-carboxylate (ACC) deaminase, and siderophores. One strain, PGR-11, was found to thrive in growth media that was supplemented with high concentrations of metalaxyl, carbendazim, and tebuconazole. Despite increasing pesticide concentrations, *Pseudomonas* sp. Continued to synthesize PGP substances. The researchers evaluated the phytotoxicity of the pesticides both in vitro and under pot-house conditions using a *Vigna radiata* (L.) crop. The results showed that increasing concentrations of chemical pesticides negatively impacted the growth, physiological and biochemical features. However, pesticide-tolerant *Pseudomonas* sp. Relieved the toxicity and improved the biological attributes of the plant. Bioinoculated plants showed significant enhancement in germination attributes, dry biomass, symbiotic features, and yield features compared to uninoculated plants. The aim of the study conducted by Huo et al. [67] was to evaluate the potential of siderophore-producing rhizobacteria in bioremediating heavy metal (HM) contamination in *Panax ginseng*. In vitro tests were conducted to assess the plant growth-promoting characteristics and HM resistance of various isolates from the ginseng rhizosphere. Based on these tests, *Mesorhizobium panacihumi*, a siderophore-producing strain, was selected as the candidate for further experiments. In planta (pot tests) and in vitro (medium tests) experiments were then conducted to investigate the capacity of the SPR candidate to alleviate oxidative stress and enhance HM resistance in *P. ginseng*. Results from the in vitro tests demonstrated that *M. panacihumi* had higher HM resistance than the other tested isolates. In the in planta studies, two-year-old ginseng seedlings exposed to a 25 mL (500 mM) Fe solution showed lower biomass and higher reactive oxygen species levels than control seedlings. However, seedlings treated with 10^8^ CFU mL^−1^ for 10 min showed increased biomass and levels of antioxidant genes and nonenzymatic antioxidant compounds compared to untreated seedlings. These findings suggest that *M. panacihumi* has the potential to enhance the growth and health of *P. ginseng* and could be used to remediate heavy metal contamination in ginseng fields. In the study conducted by Gao et al. [68], three Pseudomonas bacterial strains were evaluated, isolated from the rhizosphere of Fe-efficient apple rootstocks. Indole acetic acid-like substances and siderophores were found to be released by all three strains. In alkaline soil conditions, Fe-inefficient rootstocks treated with these *Pseudomonas* strains showed increased plant biomass, root growth, and Fe content. The production of pyoverdine, a siderophore that chelates Fe^3+^ and improves the bioavailability of Fe to plants, was observed in these bacteria. Pyoverdine was extracted from the bacterial culture supernatant and used in hydroponic trials with a Fe-deficient solution. These trials resulted in a significant reduction in chlorosis induced by Fe deficiency and an improvement in Fe absorption.

## 6. Nutrient Efficiency Units for Reducing Chemical Fertilizers

Over fifty percent of conventional nitrogen (N) fertilizer applied to cropping systems can be lost to the environment, leading to water and air pollution. To sustain crop productivity without harming the environment, it is essential to implement farming methods that ensure efficient fertilizer use. Utilizing biofertilizers with proven benefits for plant nutrition and soil health is one way to achieve this. In this context, the use of plant growth-promoting rhizobacteria (PGPR) has gained considerable attention for improving plant nutrient uptake and utilization. PGPR are known to promote plant growth through various mechanisms, such as enhancing the availability of nutrients, increasing root biomass and area, and improving the plant’s nutrient absorption capacity. Consequently, there is a growing interest in exploring the potential of PGPR to enhance plant nutrient supply and promote plant growth [69,70]. Biofertilizers containing PGPR are becoming more popular due to their economic and environmental benefits. The global market for plant growth stimulants, which include biofertilizers, is expected to grow by 12% annually [71]. In nutrient-deficient agricultural environments, particularly in the tropics, plant development is often hindered by insufficient nutrient availability. Most agricultural crops exhibit nutrient use efficiency of less than 50% in many agricultural regions, exacerbating the problem [72]. Plant growth-promoting rhizobacteria (PGPR) play a crucial role in regulating geochemical nutrient cycles and making nutrients available to plants and the soil microbial community. Incorporating these beneficial bacteria as bioinoculants can significantly increase nutrient availability in the soil, reduce reliance on chemical fertilizers, minimize environmental contamination, and promote sustainable agricultural practices [71,73].

Adopting sustainable agricultural practices that involve gradually reducing the use of synthetic agrochemicals, increasing the utilization of biowaste-derived substances, and harnessing the biological and genetic potential of crop plants and microbes is a viable strategy to combat rapid environmental degradation, ensure high agricultural productivity, and improve soil health [18]. In addition to the genetic manipulation of crop physiology and metabolism for yield enhancement, particular members of the soil microbial community, particularly those residing in the plant rhizosphere, may aid plants in preventing or partially overcoming environmental stresses. The discovery and subsequent use of biofertilizers and other microbial products, such as organic extracts and vermicompost beverages, resulted from the search for environmentally friendly alternatives to hazardous agrochemicals. These nontoxic and environmentally benign microbial products could promote plant health and growth. Rahim [74] conducted an exhaustive investigation on the effect of phosphorus on wheat and its phosphorus use efficacy. The production of grain increased significantly. Plant growth-promoting rhizobacteria (PGPR) as biofertilizers and biological control agents are a viable alternative to synthetic agrochemicals for crop production [75,76]. Bashir et al. [77] determined from their research that the administration of 100 kg P per hectare of wheat has a significant impact on wheat production. It will increase the biological productivity, plant height, number of tillers, P efficiency, harvest index, and more. Rahim [74] conducted an exhaustive investigation on the effect of phosphorus on wheat and phosphorus use efficacy. 

The production of grain has increased significantly. Continuous research and validation are necessary for the creation of new products. The products should be evaluated against various environmental conditions, including crop, climate, soil type, and agricultural practices, in order to generate ranges of potentially useful microbial products. This would result in a greater comprehension of their sustainable production potential and practicability. Figure 1 shows a schematic representation of the steps required to isolate and characterize bacteria that promote plant growth. Figure 2 shows a schematic representation comparing uninoculated and inoculated plants with bacterial endophytes and several abilities related to crop growth promotion and Figure 3 shows the modes of application of PGPR. Table 1 summarises the bacterial species, abilities, experimental conditions, and results promoted by the application in crops. 

## 7. Future Perspectives

Plant growth-promoting rhizobacteria (PGPR) are a group of beneficial soil bacteria that colonize the root surface and promote plant growth and health via multiple mechanisms. PGPR can enhance plant growth by increasing nutrient availability, producing plant growth hormones, stimulating root development, and protecting plants from diseases and parasites. Plant growth-promoting rhizobacteria (PGPR) have a bright future. These beneficial bacteria in agriculture and horticulture are gaining popularity due to their potential to improve plant growth and reduce the use of synthetic fertilizers and pesticides, and improve soil health. Future PGPR research will likely concentrate on developing new strains of bacteria that are more effective at promoting plant growth and elucidating the molecular mechanisms through which PGPR interact with plants. This could contribute to developing agricultural practices that rely less on chemical inputs and are more efficient and sustainable. The use of PGPR in biofertilizers and biopesticides is also anticipated to increase. Biofertilizers containing PGPR can improve plant growth and yield by adding nutrients to the soil and enhancing its fertility. Biopesticides containing PGPR can aid in the sustainable and environmentally benign control of plant maladies and parasites. In addition, the use of PGPR in crop breeding and genetic engineering could create cultivars that are more resistant to abiotic and biotic stresses, such as drought, salinity, and plant diseases. This could contribute to establishing more resilient and sustainable crop varieties that can withstand changing environmental conditions and contribute to food security. PGPR offer a sustainable and environmentally benign alternative to conventional agricultural practices and have the potential to contribute to developing more sustainable and resilient agriculture.

## 8. Conclusions

This review emphasizes the potential of biologically dependent instruments, specifically PGPR, to assist in addressing global food production issues. Before these tools can be applied to real-world situations, it is evident that there are significant knowledge deficits that must be filled. PGPR could be the key to sustainable crop productivity and efficient nutrient management.

## Figures and Tables

**Figure 1 microorganisms-11-01088-f001:**
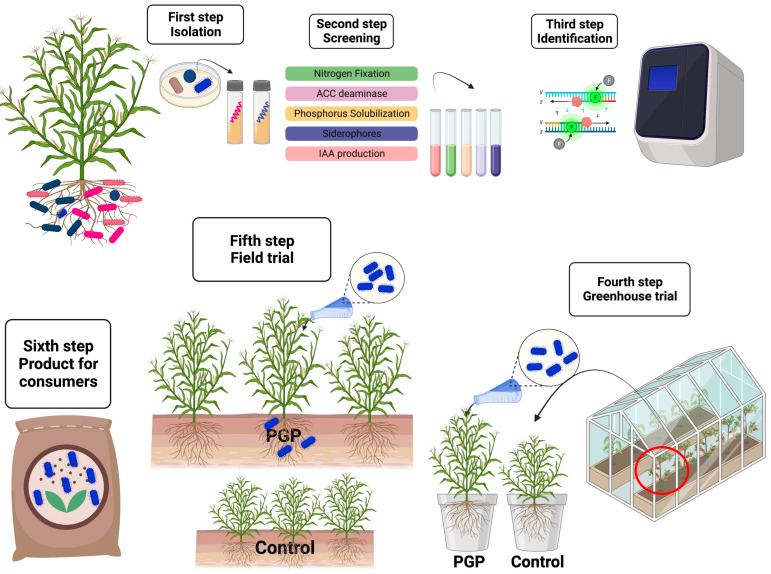
Schematic representation of the steps required to isolate and characterize bacteria that promote plant growth.

**Figure 2 microorganisms-11-01088-f002:**
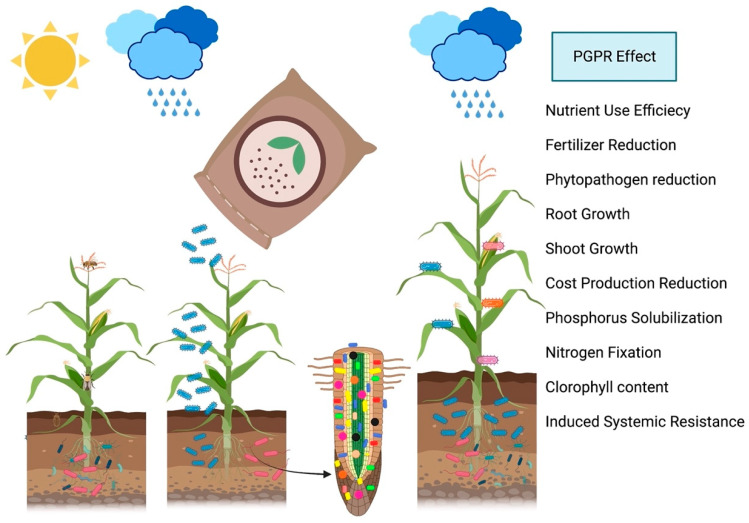
This schematic diagram compares uninoculated plants with those that have been inoculated with bacterial endophytes, illustrating several important benefits of this process for promoting plant growth in crops. These benefits include increased aerial growth, reduced susceptibility to disease, enhanced nutrient uptake, improved root growth, reduced presence of harmful phytopathogens, and induced systemic resistance. By harnessing the power of beneficial bacteria within plants, farmers can ensure healthier and more productive crops, leading to higher yields and better food security.

**Figure 3 microorganisms-11-01088-f003:**
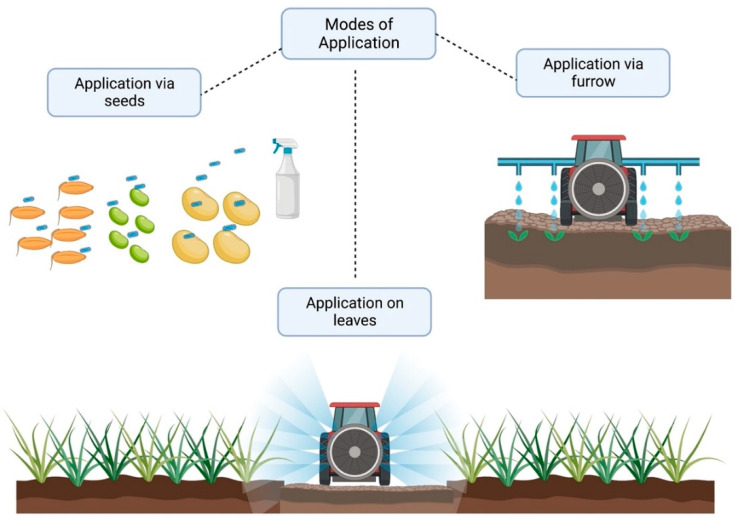
Modes of application of PGPR.

**Table 1 microorganisms-11-01088-t001:** Bacterial species, abilities, experimental conditions, and results achieved by the application in crops.

PGPR	Abilities	Condition	Results	References
*G. diazotrophicus* and *A. brasilense*	Siderophore production	Hydroponia	Increased iron in the plants	[60]
*Pseudomonas* sp.	Siderophore production and PGPR properties	Vase with vermiculite	Iron acquisition	[78]
*Lysinibacillus* sp. and *Paenibacillus dendritiformis*	Seedling protection	Maize in vase	Improved seed germination	[19]
*Bacillus subtilis*	Seedling protection	Pot experiment	Protection against *Cephalosporium maydis*	[62]
*Bacillus* sp.	IAA production Phosphorus solubilization	Greenhouse	Increased rice seedling Increase P availability	[79]
*Mesorhizobium panacihumi*	Siderophore production	Pot tests	Reduced soil contamination	[67]
*Bacillus* sp. *Citrobacter* sp. *Citrobacter* sp. *Flavobacterium* sp. and *Pantoea* sp.	Phytohormones production; siderophore; hydrogen cyanide	Greenhouse	Reduced fungal phytopathogens	[20]
*Pseudomonas* sp.	Protection against fungicide	Greenhouse	Improved cultivation of radish plants	[65]
*Pseudomonas fluorescens Rhizobium leguminosarum Bacillus brevis* *Azotobacter vinelandii*	Siderophore production	Chamber	Protection against organophosphate pesticides	[80]
*Anabaena azotica*	Nitrogen fixation	Field experiment	Chemical nitrogen reduction	[41]
*Brucella* sp. and *Pseudomonas brassicae*	Siderophore production	Field experiment	High production in iron-deficient soil	[81]
*Bacillus aryabhattai*	Siderophore production	Field experiment	Increased production in iron-deficient soil	[59]
*Rhodopseudomonas palustris*	Phosphorus solubilization	Greenhouse	Improved soil fertility	[82]

## Data Availability

Not applicable.

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
