# Peer review of "Plant Growth-Promoting Rhizobacteria for Sustainable Agricultural Production"

_microorganisms, 2023, doi:10.3390/microorganisms11041088_

Round 1
Reviewer 1 Report
Manuscript microorganisms-2345579 entitled “Plant Growth-Promoting Microorganisms for Sustainable Agricultural Production”. Please notice the following:
General view: The manuscript illustrated a topic of Plant Growth-Promoting Rhizobacteria (PGPR). The topic is very interesting.This problem is relevant for journal scope.
The Autors expressed their idea in moderate language and grammar. The manuscript might require copyediting and proofreading up to a little degree to provide more simplified sentences.
The introduction is easy detailed. The concept and aim are clearly defined. The presentation and discussion of the presented topicis clear and very detailed.
Suggests supplementing the "Introduction" with information bringing a new scientific contribution. Please provide a review of the literature in this area.The text can be supplemented with information on new unconventional fertilizers.
To raise the level of paper, please use the articles on fertilizers:
doi: 10.3390/en16041747
doi: 10.3390/agriculture11111121
It is worth quoting other publications on PGPMs here, e.g:
doi: 10.3390/metabo13020247
In general, the paper is well written.
The literature review is conclusive and of interest for this topic. I have not found any important formal mistakes or typo errors, but you need to do some organization in the points to improve readability. Formally, the paper is well written and easy to understand.
Please cite more papers from MDPI journals at the last 2-3 years in the similar topic of this research.
Weaknesses to be corrected:
1. Keywords should be in alphabetical order.
2. Divide the manuscript into sections and subsections. It will be better to read this paper.
3. Conclusions should be shorter. Give some of your most important thoughts about the literature review
4. Table 1 should be deleted from the conclusions. Please move it to another section.
5. Not all abbreviations are explained, e.g. HCN, PS3, AZ2- should be completed
6. Correct the units, check the indexes, e.g. Line 300, Line 315, Line 316
The manuscript follows the formal regulations of MDPI journals.
I suggest the acceptance after minor revision
Author Response
Reviewer1 - Manuscript microorganisms-2345579 entitled “Plant Growth-Promoting Microorganisms for Sustainable Agricultural Production”. Please notice the following:
General view: The manuscript illustrated a topic of Plant Growth-Promoting Rhizobacteria (PGPR). The topic is very interesting. This problem is relevant for journal scope.
Answer: We authors, thank the reviewer for the opportunity given to us to improve our manuscript. We really appreciate your work.
All the modifications were written in red color
Reviewer1 -The Autors expressed their idea in moderate language and grammar. The manuscript might require copyediting and proofreading up to a little degree to provide more simplified sentences.
Answer: It has been done.
Reviewer1 -The introduction is easy detailed. The concept and aim are clearly defined. The presentation and discussion of the presented topicis clear and very detailed.
Answer: Thank you
Reviewer1 - Suggests supplementing the "Introduction" with information bringing a new scientific contribution. Please provide a review of the literature in this area. The text can be supplemented with information on new unconventional fertilizers.
Answer: All the manuscript suggested below have been cited
Reviewer1 - To raise the level of paper, please use the articles on fertilizers:
doi: 10.3390/en16041747
doi: 10.3390/agriculture11111121
It is worth quoting other publications on PGPMs here, e.g:
doi: 10.3390/metabo13020247
In general, the paper is well written.
Reviewer1 -The literature review is conclusive and of interest for this topic. I have not found any important formal mistakes or typo errors, but you need to do some organization in the points to improve readability. Formally, the paper is well written and easy to understand.
Answer: Thank you, reviewer.
Reviewer1 - Please cite more papers from MDPI journals at the last 2-3 years in the similar topic of this research.
Answer: More studies from MDPI have been cited.
Weaknesses to be corrected:
- Keywords should be in alphabetical order.
Answer: It has been done.
- Divide the manuscript into sections and subsections. It will be better to read this paper.
Answer: It has been done.
- Conclusions should be shorter. Give some of your most important thoughts about the literature review
Answer: It has been shortened.
- Table 1 should be deleted from the conclusions. Please move it to another section.
Answer: It has been moved.
- Not all abbreviations are explained, e.g. HCN, PS3, AZ2- should be completed
Answer: The abbreviations have been explained or deleted.
- Correct the units, check the indexes, e.g. Line 300, Line 315, Line 316
Answer: These units were removed.
The manuscript follows the formal regulations of MDPI journals.
I suggest the acceptance after minor revision
Answer: The authors thank the reviewer.

Reviewer 2 Report
Microorganisms
Manuscript ID: 2345579
Title: Plant growth-promoting microorganisms for sustainable agricultural production
The manuscript review studies related to the characterization of plant growth-promoting microorganisms, in specific those capable of improving nutrients availability (P, N, and Fe), helping plants to counteract pests, diseases, abiotic stress, as well as phytohormone production capability, altogether improving plant/crops growth and productivity.
The topic is interesting and overall the manuscript is well redacted, however in the last year several reviews related to plant growth-promoting microorganisms have been published in different journals, but the authors didn´t make reference to similar reviews, in special they not contrasted the similitudes and differences between the current manuscript and all previously published reviews in the topic. These aspects are worthy of mention/discussion in the abstract and introduction; highlighting the novel aspects of the present manuscript.
Additional comments:
Lines 14-15, review the redaction of the fragment, it is repetitively, aspects related to heavy metals are presented as independent, when are all related.
Lines 22-23, the fragment “nutrient efficiency units for reducing chemical fertilizers for sustainable agriculture”, is not clear in the context of the redaction, check the redaction or eliminate it.
Line 62, eliminate “and” between “organic and energetic”
Line 72, eliminate extra space in “genome. However”
Lines 84-86, review the redaction, if shoots and roots development is promoted, hence plant growth is promoted
Line 91, add a space in “productivity[15]”
Line 115, eliminate “and”
Line 118, change “their” for “the”
Line 118, define the acronym “NUE”, or eliminate it if not used next in the manuscript
Line 126, change the position of the acronym “BNF” next to “Biological nitrogen fixation”
Line 152, add a space in “Pandey et al.[29]”
Line 156, change first “and” by a “comma”
Line 164, add a space in “Wan et al.[33]”
Line 168, eliminate comma and xtra space in “Liu et al., [34] have”
Line 171, Line 164, add a space in “Pantigoso et al.[35]”
Line 179, eliminate extra space in “Staphylococcus ,”
Line 195, eliminate extra period in “soils. [37]. This”
Line 211, correct super index in (Fe3+)
Line 216, add a space in “Sultana et al.[44]”
Line 243 and 253, check the formant in “Fe(III) to Fe(II)”, previously Fe3+ and Fe 2+ were used
Lines 266-269, use italic for all the bacterial species names
Line 300, correct super index in (kg-1)
Line 310, previously PGPR was used instead “plant growth-promoting (PGP) rhizobacterial”
Line 315 and 316, correct super index in (L-1)
Lines 316-318, use italic for all the species names
Lina 318, eliminate extra period in “crops..”
Lines 323-324, add a space in “Huo et al.[52]”
Line 329, use italic for “in vitro”
Lines 323-324, add a space in “problem[56].”
Line 370, eliminate extra period in “practices. [55].”
Line 382, eliminate extra space in “Rahim [57] conducted”
Line 387, add a space in “Bashir et al.[60]”
In Table 1, use numbers for references, and check the format of “sp”, it is in italics or normal font, with and without a period, check all and adequate for homogeneity.
Review and adequate the references format according to the journal requirements
English quality is adequate, with only minor aspects in redaction, as were mentioned above in the comments and suggestions for authors.
Author Response
Reviewer2 - The topic is interesting and overall the manuscript is well redacted, however in the last year several reviews related to plant growth-promoting microorganisms have been published in different journals, but the authors didn´t make reference to similar reviews, in special they not contrasted the similitudes and differences between the current manuscript and all previously published reviews in the topic. These aspects are worthy of mention/discussion in the abstract and introduction; highlighting the novel aspects of the present manuscript.
Answer: The novelty brought from this review is: that this review brings the studies that used PGPR for sustainable production in a practical way, such as bacteria used for reducing phosphorus fertilizer, bacteria used for reducing nitrogen fertilizer, and bacteria used for improving fertilizer uptake. It has been added to the abstract.
Additional comments:
Reviewer2 - Lines 14-15, review the redaction of the fragment, it is repetitively, aspects related to heavy metals are presented as independent, when are all related.
Answer: It has been corrected accordingly
Reviewer2 - Lines 22-23, the fragment “nutrient efficiency units for reducing chemical fertilizers for sustainable agriculture”, is not clear in the context of the redaction, check the redaction or eliminate it.
Answer: This fragment has been deleted.
Reviewer2 - Line 62, eliminate “and” between “organic and energetic”
Answer: It has been eliminated.
Reviewer2 - Line 72, eliminate extra space in “genome. However”
Answer: It has been eliminated.
Reviewer2 - Lines 84-86, review the redaction, if shoots and roots development is promoted, hence plant growth is promoted
Answer: It has been changed.
Reviewer2 -Line 91, add a space in “productivity[15]”
Answer: It has been done.
Reviewer2 -Line 115, eliminate “and”
Answer: It has been eliminated.
Reviewer2 -Line 118, change “their” for “the”
Answer: It has been changed.
Reviewer2 -Line 118, define the acronym “NUE”, or eliminate it if not used next in the manuscript
Answer: It has been deleted.
Reviewer2 -Line 126, change the position of the acronym “BNF” next to “Biological nitrogen fixation”
Answer: It has been changed.
Reviewer2 -Line 152, add a space in “Pandey et al.[29]”
Answer: It has been changed.
Reviewer2 -Line 156, change first “and” by a “comma”
Answer: It has been changed.
Reviewer2 -Line 164, add a space in “Wan et al.[33]”
Answer: It has been done.
Reviewer2 -Line 168, eliminate comma and xtra space in “Liu et al., [34] have”
Answer: It has been eliminated.
Reviewer2 -Line 171, Line 164, add a space in “Pantigoso et al.[35]”
Answer: It has been done
Reviewer2 -Line 179, eliminate extra space in “Staphylococcus ,”
Answer: It has been done.
Reviewer2 -Line 195, eliminate extra period in “soils. [37]. This”
Answer: It has been done.
Reviewer2 -Line 211, correct super index in (Fe3+)
Answer: It has been done.
Reviewer2 -Line 216, add a space in “Sultana et al.[44]”
Answer: It has been done.
Reviewer2 -Line 243 and 253, check the formant in “Fe(III) to Fe(II)”, previously Fe3+ and Fe 2+ were used
Answer: It has been done.
Reviewer2 -Lines 266-269, use italic for all the bacterial species names
Answer: It has been done
Reviewer2 -Line 300, correct super index in (kg-1)
Answer:
Reviewer2 -Line 310, previously PGPR was used instead “plant growth-promoting (PGP) rhizobacterial”
Reviewer2 -Line 315 and 316, correct super index in (L-1)
Answer: It has been done
Reviewer2 -Lines 316-318, use italic for all the species names
Answer: It has been done.
Reviewer2 -Lina 318, eliminate extra period in “crops..”
Answer: It has been done
Reviewer 2 -Lines 323-324, add a space in “Huo et al.[52]”
Answer: It has been done
Reviewer2 -Line 329, use italic for “in vitro”
Answer: It has been done
Reviewer2 -Lines 323-324, add a space in “problem[56].”
Answer: It has been changed.
Reviewer2 -Line 370, eliminate extra period in “practices. [55].”
Answer: It has been done.
Reviewer2 -Line 382, eliminate extra space in “Rahim [57] conducted”
Answer: It has been done.
Reviewer2 -Line 387, add a space in “Bashir et al.[60]”
Answer: It has been done.
Reviewer2 -In Table 1, use numbers for references, and check the format of “sp”, it is in italics or normal font, with and without a period, check all and adequate for homogeneity.
Answer: It has been done.
Review and adequate the references format according to the journal requirements

Reviewer 3 Report
Comment sheet
The research article entitled ‘Plant Growth- Promoting Microorganisms for Sustainable Agricultural Production’ withholds importance in the field of growth promotion study, biological control of plant diseases. I would recommend this research article to accept after minor revision and concerns stated below.
1. I have concern in title, Authors wrote the title indicating ‘Microorganisms; However, they only described about rhizobacteria and seed endophytes. I believe, if authors write the title as ‘Microorganisms’, there would be other microbes should authors addressed in this review article.
2. Authors need to generalize the PGPB in details before other subtopics.
3. I could not see review regarding the mechanisms of PGPR (antibiotic gene production, SAR, ISR). Authors need to review those aspects too.
4. Seed microbiome and rhizosphere colonization can be separately reviewed.
5. In fig. 1. Authors need to provide tentative time frame for first to fifth step. Since, utilizing PGPB in field in commercially is not easy step. Authors need to provide more detail insights, regarding each step.
6. As we know that, beneficial microorganisms can be applied to plant in various ways. Please write more in detail how these PGPR can be applied to plants graphically.
7. In table 1, authors must provide details of endophytes, and other different types of beneficial microorganisms not only bacteria.
8. Authors need to check the format to write the scientific name of microbes in italic through out the text.

Moderate editing of English language is needed
Author Response
Reviewer3- Comment sheet
The research article entitled ‘Plant Growth- Promoting Microorganisms for Sustainable Agricultural Production’ withholds importance in the field of growth promotion study, biological control of plant diseases. I would recommend this research article to accept after minor revision and concerns stated below.
Answer: We authors thank the reviewer for the opportunity given to us to improve our manuscript. We really appreciate your work.
All the modifications were written in red color
Reviewer3- I have concern in title, Authors wrote the title indicating ‘Microorganisms; However, they only described about rhizobacteria and seed endophytes. I believe, if authors write the title as ‘Microorganisms’, there would be other microbes should authors addressed in this review article.
Answer: We agree with the reviewer. The title has been changed for Plant Growth-Promoting Rhizobacteria for Sustainable Agricultural Production
Reviewer3- Authors need to generalize the PGPB in details before other subtopics.
Answer: It has been done.
Reviewer3- I could not see review regarding the mechanisms of PGPR (antibiotic gene production, SAR, ISR). Authors need to review those aspects too.
Answer: It has been addressed.
Reviewer3- Seed microbiome and rhizosphere colonization can be separately reviewed.
Answer: Both are in separate sections.
Reviewer3- In fig. 1. Authors need to provide tentative time frame for first to fifth step. Since, utilizing PGPB in field in commercially is not easy step. Authors need to provide more detail insights, regarding each step.
Answer; More details have been added.
Reviewer3- As we know that, beneficial microorganisms can be applied to plant in various ways. Please write more in detail how these PGPR can be applied to plants graphically.
Answer: It has been done.
Reviewer3- In table 1, authors must provide details of endophytes, and other different types of beneficial microorganisms not only bacteria.
Answer: We decided to change the title to Rhizobacteria. Therefore, we decided to cite only bacteria in the table 1.
Reviewer3 - Authors need to check the format to write the scientific name of microbes in italic through out the text.
Answer: It has been checked.
Reviewer3 : Comments on the Quality of English Language
Moderate editing of English language is needed
Answer: It has been checked.

Round 2
Reviewer 2 Report
The authors addressed the reviewer's comments, and no additional issues were identified in the manuscript.